# Comparative Analysis of Melatonin and Polydeoxyribonucleotide: Possible Benefits of Co-Treatment Effects and Potential Synergistic Applicability

**DOI:** 10.3390/ijms26125703

**Published:** 2025-06-13

**Authors:** Su Kil Jang, Jaeseok Choi, Hye Won Lim, Hong-Gyum Kim, Yeong-Min Yoo

**Affiliations:** 1Shin Sung Bio Pharm Inc., Saimdang-ro 641-22, Gangneung 25451, Republic of Korea; waterroad79@daum.net; 2Institute of Environmental Research, Kangwon National University, Chuncheon 24341, Republic of Korea; gobiobotia@kangwon.ac.kr; 3Novel Vita Inc., Dongnae-myeon, Chuncheon 24398, Republic of Korea; novelvita@naver.com (H.W.L.); adonia@empas.com (H.-G.K.)

**Keywords:** melatonin, PDRN, potential synergistic applicability, clinical applicability

## Abstract

This paper explores the enhancement of pharmacological outcomes through the combined use of melatonin and polydeoxyribonucleotide (PDRN), hypothesizing that their simultaneous application might surpass the effectiveness of individual use. Melatonin is a hormone that modulates sleep, oxidative stress and inflammation, and exerts analgesic and anti-inflammatory effects. Conversely, PDRN is well-known for its significant contributions to tissue regeneration and its role in promoting angiogenesis. This article details the pharmacological effects and mechanisms of each compound, suggesting that their integration could amplify their individual benefits, particularly in the realms of wound healing and various medical applications. This paper seeks to provide a comprehensive analysis of the interactions between melatonin and PDRN by reviewing existing studies, thereby paving the way for novel therapeutic strategies. It emphasizes the need for further clinical trials and research to optimize the use of this combination for the improved treatment of diverse cellular or tissue conditions. In conclusion, further research is needed to optimize combination therapies involving melatonin and PDRN, with the goal of confirming their enhanced benefits when used together. In conclusion, further research is necessary to optimize combination therapies involving melatonin and PDRN to confirm their enhanced benefits when used in conjunction. This review emphasizes the importance of exploring their potential synergistic effects and developing effective therapeutic strategies across various medical disciplines.

## 1. Introduction

Employing multiple medications in tandem as an effective therapeutic strategy is advantageous for disease management. This approach can yield superior outcomes compared to monotherapy, potentially offering a reduction in side effects, a decreased likelihood of treatment failure, and mitigating the necessity for the development of new medications [1,2]. Besides regulating sleep, melatonin has demonstrated capabilities in modulating oxidative stress and inflammation and offers potential therapeutic benefits for neurodegenerative diseases and cancer [3,4]. Polydeoxyribonucleotide (PDRN) promotes tissue regeneration, mitigates inflammation through its anti-inflammatory properties, and stimulates angiogenesis via vascular endothelial growth factor (VEGF) production [5,6,7]. Therefore, the synergistic effects of melatonin and PDRN hold considerable promise in pharmacological research, especially in enhancing wound healing and tissue regeneration, and expanding to a broader spectrum of medical applications.

Therefore, the distinct mechanisms of action and varied pharmacological effects of each biomolecule are reviewed to compare and analyze the therapeutic potential of melatonin and PDRN. Importantly, we have placed special emphasis on the synergistic effects observed when these two substances are combined. This review elucidates how co-administering melatonin and PDRN could enhance the benefits of each, pave the way for novel therapeutic approaches, and ultimately lay the groundwork for developing new and effective treatments across various applications. By reviewing existing studies and exploring potential effects, this review aimed to provide a comprehensive understanding of the interactions between melatonin and PDRN. The papers utilized in this review were predominantly recent publications from 2020 to 2025, selected through a comprehensive PubMed search using ‘melatonin’ and ‘PDRN’ as primary keywords, along with additional terms such as ‘receptor-mediated actions’, ‘antioxidant activity’, and ‘anti-inflammatory effects’, among others.

## 2. Mechanisms of Action and Effects of Melatonin

### 2.1. Melatonin: A Well-Studied and Versatile Regulatory Molecule

Melatonin, also known as N-acetyl-5-methoxytryptamine, is an intriguing hormone naturally present in most organisms and plays a critical role in regulating numerous physiological processes. The synthesis of melatonin predominantly takes place in the pineal gland—a small, pinecone-shaped endocrine gland situated deep within the brain. In the pineal gland, a complex cascade of enzymatic reactions transforms tryptophan into serotonin, which subsequently serves as a precursor for melatonin [8,9]. Once produced, melatonin is directly released into the bloodstream and cerebrospinal fluid, from where it circulates throughout the body and elicits various effects on target tissues and organs [8,9,10].

The release of melatonin is not constant and is intricately linked to the light–dark cycle [10,11,12]. Specifically, melatonin production increases in response to darkness, while its levels naturally decrease as daylight emerges. This rhythmic production of melatonin over the course of the day and night is vital for the regulation of the circadian rhythm, commonly referred to as the body’s internal biological clock. This clock not only governs the sleep–wake cycle, but also influences other physiological functions, including hormone secretion, body temperature, and appetite. Although other tissues like bone marrow and the gut can produce melatonin, the pineal gland remains the principal site for melatonin synthesis and the regulation of circadian rhythms [10,11,12].

Melatonin has been extensively studied and is recognized as a highly versatile regulatory molecule with a broad spectrum of physiological roles [10,11]. Unlike other hormones that have more restricted functions, melatonin exerts its influence through multiple mechanisms, demonstrating its extensive impact on the body. These mechanisms encompass direct antioxidant activities, receptor-mediated signaling, and the modulation of enzymatic activity. Furthermore, melatonin affects gene expression and interacts with various signaling pathways, underscoring its significant and multifaceted regulatory functions. Among these mechanisms, its primary action as a regulatory hormone for the sleep–wake cycle is facilitated by its binding to specific receptors, MT1 and MT2, which are predominantly located in the hypothalamic suprachiasmatic nucleus (SCN)—the central circadian clock of the brain. Upon binding to these receptors in the SCN, melatonin promotes sleepiness and regulates the timing of the circadian rhythm [10,11,12]. In addition to its receptor binding, melatonin also exhibits antioxidant properties. It scavenges free radicals and reduces oxidative stress, which may contribute to cellular damage and age-related decline [10,11]. Moreover, it might impact immune function and possess anti-inflammatory effects, though the detailed mechanisms behind these effects are still being actively explored. Melatonin’s influence extends beyond the brain, as it is also synthesized in other tissues, including the gut, where it may help regulate gastrointestinal function. Furthermore, melatonin can modulate the release of various hormones, such as cortisol, aiding in the synchronization of hormonal rhythms with the sleep–wake cycle [10,11].

### 2.2. Melatonin as a Medicine or Dietary Supplement

Melatonin is widely utilized as a dietary supplement and therapeutic agent, primarily for its role in regulating circadian rhythms, making it a popular choice for managing sleep disorders. Additionally, emerging research points to its broader applications. In the realm of circadian rhythm disorders, melatonin is effective in treating conditions such as delayed sleep–wake phase disorder and jet lag. Clinical guidelines advocate for the timed administration of melatonin in these instances, although its effectiveness for treating insomnia remains uncertain due to limited evidence [13,14,15].

Meta-analyses present mixed results for insomnia; however, melatonin has been shown to enhance subjective sleep quality in specific patient groups, including those with traumatic brain injury, Parkinson’s disease, and Alzheimer’s disease [16]. Significant variability in melatonin content among commercial supplements has been documented, with discrepancies ranging from −83% to +478% of the labeled amounts. Furthermore, 26% of products tested were found to contain serotonin, raising serious concerns about both quality and safety [13,14,15].

Melatonin also exhibits potential in reducing Alzheimer’s disease pathology by decreasing amyloid-beta and tau protein burdens, bolstering antioxidant defenses, and improving circadian regulation. Yet, these promising effects necessitate validation through additional clinical trials [16]. Moreover, melatonin may enhance the body’s inherent DNA repair mechanisms through its antioxidant activities, potentially mitigating cellular damage associated with disrupted circadian rhythms or poor sleep quality [17,18]. Case studies indicate that melatonin can induce growth hormone secretion, which is an effect that reverses upon cessation. While this finding may have clinical relevance, it requires further exploration [19].

Advancements in biotechnology have facilitated the development of plant-derived melatonin (phytomelatonin) and genetically engineered microorganisms for melatonin synthesis. These innovations aim to minimize the chemical by-products associated with synthetic melatonin production and respond to consumer demand for “natural” products [20].

Synthesis of melatonin is not confined to the pineal gland in vertebrates; in most other tissues and organisms, melatonin production is non-circadian. Insects provide pivotal insights for unraveling the intertwined histories and functions of melatonin and vitamin D in evolution and modern biology [21]. As a dietary supplement or medicine, melatonin is valued for its antioxidant properties and its ability to support cellular health. Notably, melatonin and its metabolites, such as those produced by UV-induced phototransformation, exhibit diverse biological activities, including cytoprotection and photoprotection [21]. These properties are evolutionarily conserved and observed across taxa, including insects, highlighting melatonin’s fundamental biological significance and its therapeutic potential as a supplement or medicinal agent. 

The antioxidant properties of melatonin have piqued interest in its potential roles in immune activation, anti-stress effects, and anti-aging therapies, although these areas remain relatively underexplored [20,22]. While it is generally regarded as safe for short-term use, the long-term effects of melatonin supplementation have yet to be thoroughly investigated. Dependency on supplements without proper medical consultation could present health risks [19]. Regulatory oversight also varies significantly by region; for instance, melatonin is sold over the counter in the US but requires a prescription in some countries, such as the UK [17,18,19]. As melatonin continues to gain traction as both a dietary supplement and therapeutic agent well beyond just sleep regulation, the inconsistent quality of supplements and the paucity of data on long-term safety underscore the urgent need for further research and regulatory enhancements.

### 2.3. Mechanisms of Action and Effects of Melatonin

#### 2.3.1. Receptor-Mediated Actions

Melatonin primarily affects various physiological processes through its interactions with high-affinity G protein-coupled receptors, specifically MT1 and MT2. These receptors are strategically positioned across the body, including in the brain (especially the SCN, which is the principal circadian pacemaker), retina, and peripheral tissues. This wide distribution enables melatonin to regulate an extensive range of functions. Upon binding to MT1 and MT2 receptors, melatonin triggers a series of intracellular signaling pathways that ultimately influence cellular activity [12,23,24,25].

An important consequence of the activation of MT1 and MT2 is the modulation of cyclic adenosine monophosphate (cAMP) levels [12,23,24,25,26,27]. In many cell types, the activation of these receptors leads to a reduction in cAMP production. This is due to MT1 and MT2 typically being coupled to Gi/o proteins, which inhibit adenylyl cyclase, the enzyme responsible for converting ATP into cAMP [12,23,24,25,26,27]. By lowering cAMP, melatonin can impact various downstream processes, including gene expression, ion channel activity, and neurotransmitter release [12,23,24,25,26,27]. Conversely, in certain cellular contexts, melatonin signaling might unexpectedly increase cAMP levels, which underscores the context-dependent nature of its effects [12,23,24,25,26,27].

Beyond influencing cAMP, activation of melatonin receptors can also affect cyclic guanosine monophosphate (cGMP) levels, although these mechanisms are less clearly defined and may involve indirect pathways. Variations in cGMP may affect smooth muscle relaxation, visual processing, and neuronal signaling [12,23,24,25,26,27,28]. Additionally, melatonin signaling can alter intracellular calcium concentrations. In certain cell types, activation of MT1 and MT2 may lead to a reduction in intracellular calcium, possibly through the inhibition of voltage-gated calcium channels or the stimulation of calcium efflux pumps [12,23,24,25,26,27,28]. Conversely, in other cell types, melatonin might transiently raise intracellular calcium levels. Such fluctuations in calcium are crucial for regulating neuronal excitability, synaptic plasticity, and other cellular functions [12,23,24,25,26,27,28].

Moreover, melatonin’s activation of MT1 and MT2 receptors can modulate the activity of various transcription factors. By influencing cAMP, cGMP, and calcium signaling pathways, melatonin indirectly impacts the phosphorylation and nuclear translocation of transcription factors such as cAMP response element-binding protein (CREB) and nuclear factor-kappa B (NF-κB) [12,23,24,25,26,27,28,29]. This modulation alters gene expression patterns, influencing the synthesis of proteins relevant to circadian rhythms, immune function, antioxidant defenses, and other essential processes [12,23,24,25,26,27,28,29]. The specific transcriptional targets and subsequent phenotypic effects vary with the cell type and biological context [12,23,24,25,26,27,28,29].

Recent studies have suggested that melatonin and its metabolites can act as ligands for the aryl hydrocarbon receptor (AhR) and peroxisome proliferator-activated receptor γ (PPARγ) [30,31]. Melatonin exhibits similar potency to indole acetic acid, the natural ligand of AhR, and activation of AhR promotes transcriptional activation of the CYP1A1 promoter, as well as translocation of the AhR protein from the cytoplasm to the nucleus in human keratinocytes [30,31]. PPARγ binds to its ligand-binding domain and stimulates the transcriptional activity of the PPAR-responsive element promoter [30,31]. These findings suggest that melatonin and its metabolites exert their photoprotection, anti-aging, anti-cancer, and antioxidant effects through interaction with AhR and PPARγ, via a mechanism independent of MT1 and MT2 receptors. Furthermore, they indicate that melatonin or its metabolites may be useful for preventing and treating skin diseases and skin aging.

#### 2.3.2. Antioxidant Activity

Melatonin is an exceptional scavenger of reactive oxygen and nitrogen species, including hydroxyl radicals (·OH), hydrogen peroxide (H_2_O_2_), singlet oxygen (^1^O_2_), nitric oxide (NO), and peroxynitrite anions (ONOO^−^). Its indole group serves as the reactive center, while its methoxy and amide side chains enhance its antioxidant properties. Distinct from conventional antioxidants such as vitamins C and E, melatonin exhibits no pro-oxidant activity and demonstrates a ‘free radical scavenging cascade’, enabling one molecule to neutralize multiple reactive species [32].

Additionally, melatonin metabolites, such as 6-hydroxymelatonin and 4-hydroxymelatonin, further augment its antioxidant effects by detoxifying free radicals and related reactive compounds [33,34]. Melatonin stimulates antioxidant enzymes such as superoxide dismutase (SOD), catalase (CAT), and glutathione peroxidase (GPx), providing a robust first line of defense against oxidative damage and reducing lipid peroxidation and cellular mortality [35,36]. It further promotes the synthesis of intracellular antioxidants like glutathione (GSH) through induction of γ-glutamylcysteine synthetase activity [36].

Melatonin repairs oxidized DNA through electron transfer mechanisms and enhances DNA repair pathways. Additionally, it regenerates other antioxidants, such as glutathione and ascorbic acid, through similar electron transfer processes [34]. In animal models, melatonin has been shown to reduce levels of malondialdehyde (MDA), a marker of lipid peroxidation, and to improve tissue quality by reducing oxidative damage. Its amphiphilic nature facilitates action across diverse cellular compartments such as membranes, cytoplasm, nucleus, and mitochondria [35]. Melatonin’s antioxidant effects also improve gut health by modulating the gut microbiota composition and alleviating oxidative stress in gastrointestinal tissues [34].

Clinical trials are investigating the efficacy of melatonin in treating oxidative stress-related diseases including Alzheimer’s, Parkinson’s, stroke, and heart disease, highlighting its superior antioxidant potency relative to vitamins C and E [32]. Its capacity to prevent oxidative damage without pro-oxidative effects positions it as a viable candidate for long-term therapeutic applications [33]. Melatonin interacts synergistically with other antioxidants, enhancing its protective effects against mitochondrial electron leakage and oxidative tissue damage [33]. Recent studies have confirmed melatonin’s multifaceted role as a potent antioxidant, with applications spanning animal models, plant systems, and potential clinical settings. Its unique properties, such as amphiphilicity, cascade scavenging reactions, and enzymatic enhancement, set it apart from traditional antioxidants.

#### 2.3.3. Anti-Inflammatory Effects

Melatonin has exhibited significant anti-inflammatory properties through several mechanisms, including the modulation of inflammatory cytokines, inhibition of oxidative stress, and regulation of signaling pathways such as NF-κB and NLRP3 inflammasome activation. A systematic review and meta-analysis demonstrated that exogenous melatonin significantly lowers levels of pro-inflammatory cytokines such as interleukin-1 beta (IL-1β), interleukin-6 (IL-6), and interleukin-8 (IL-8), as well as tumor necrosis factor-alpha (TNF-α) [37]. However, it showed no significant effect on C-reactive protein (CRP), emphasizing its potential as an adjunct treatment for chronic inflammatory conditions [37].

Melatonin inhibits the nuclear translocation of NF-κB, thus preventing the transcription and translation of inflammatory cytokines such as IL-1 and TNF. Additionally, it downregulates 5-lipoxygenase expression and activates antioxidant defenses, including SOD and catalase, which reduce oxidative stress [37,38]. In models of placental inflammation, melatonin inhibits NLRP3 inflammasome activation by reducing ROS and enhancing Nrf2, a crucial antioxidant protein [38]. This suppression of prolonged inflammation is achieved through the reduction in oxidative stress and cytokine production [37].

Melatonin’s anti-inflammatory effects have been explored in secondary traumatic brain injury (TBI) models, where it mitigates neuroinflammation by reducing oxidative damage and inflammatory cytokines [39]. Additionally, melatonin shows promise in alleviating oxidative stress-induced complications such as preeclampsia by maintaining the ROS-antioxidant balance in the placenta [38]. Infection-induced airway inflammation has also been reduced in animal models treated with melatonin, indicating its potential utility in managing respiratory conditions [40]. Melatonin is generally well tolerated with minimal side effects, making it a viable option for long-term management of chronic inflammation [37,39]. While animal studies robustly support melatonin’s anti-inflammatory capabilities, further human clinical trials are necessary to verify its effectiveness across diverse populations and conditions. Future research should also aim to determine the optimal dosages and administration routes for specific inflammatory disorders [37,38,39].

#### 2.3.4. Modulation of Inflammasome Activity

Recent studies have underlined melatonin’s ability to modulate inflammasome activity, particularly the NLRP3 inflammasome, through several mechanisms. Melatonin disrupts the IL-1β/NF-κB-NLRP3 inflammasome positive feedback loop both in vitro and in vivo, by suppressing NF-κB signaling and reducing mitochondrial ROS production. This action helps alleviate inflammation and slow the progression of intervertebral disk disease (IVDD) [41,42]. In models with high-fat diets, melatonin inhibited NLRP3 inflammasome activation via TLR4/NF-κB and P2X7 pathways, demonstrating its potential to alleviate liver inflammation [43]. Moreover, melatonin reduced NLRP3 inflammasome activation in hippocampal microglia exposed to dim blue light at night, mediated by the MT2 receptor which downregulated NF-κB and NLRP3 expression, thus mitigating neuroinflammation and neuronal damage [44].

Melatonin also reduces oxidative stress, a critical activator of NLRP3, by promoting autophagic flux, a process partly mediated through α7 nicotinic acetylcholine receptors (α7 nAChRs). In α7 nAChR knockout mice, the anti-inflammatory effects of melatonin were notably diminished, underscoring the receptor’s role in modulating autophagy and oxidative stress [45].

In amyloid beta-induced inflammation models, melatonin attenuated inflammasome-associated mechanisms, protecting against neurodegenerative processes [46]. Melatonin consistently downregulates NF-κB signaling across various models, including IVDD, neuroinflammation, and liver inflammation. This suppression diminishes the transcription of pro-inflammatory cytokines, such as IL-1β and TNF-α, thereby inhibiting NLRP3 priming and activation [41,43,44]. Melatonin’s interaction with its MT2 receptor is pivotal in mitigating inflammatory responses through the inhibition of NF-κB and NLRP3 activation in microglia [44]. Additionally, melatonin’s influence on the NLRP3 inflammasome has been linked to diminished vascular inflammation related to atherosclerosis [47]. In models exhibiting cognitive impairment triggered by factors such as dim blue light or sleep deprivation, melatonin has been shown to reduce hippocampal NF-κB activity and pro-inflammatory cytokine production [44]. Melatonin exhibits significant anti-inflammatory effects by modulating the NLRP3 inflammasome through multiple mechanisms: suppression of NF-κB signaling, reduction in oxidative stress through autophagy, and receptor-mediated pathways (e.g., MT2 and α7 nAChRs) [44,45,46,47]. These findings underscore its potential as a therapeutic agent in conditions such as IVDD, neuroinflammation, liver diseases like NASH, and cognitive decline.

#### 2.3.5. Analgesic Properties

Melatonin is increasingly being studied for its potential analgesic properties, with research highlighting its role in managing neuropathic, inflammatory, and chronic pain. The analgesic effects of melatonin are attributed to its interactions with melatonin receptors (MT1 and MT2), and its modulation of oxidative stress and inflammation. MT2 receptors, in particular, are crucial in pain modulation, as demonstrated in animal models where melatonin alleviated neuropathic pain in a dose-dependent manner [48,49]. The nitric oxide pathway and the opioid receptor systems may also contribute. For instance, pretreatment with L-arginine reversed the protective effects of melatonin in neuropathic pain models, suggesting an involvement of nitric oxide [48]. 

A meta-analysis conducted in 2020 found that melatonin significantly reduced the intensity of chronic pain, particularly in conditions such as fibromyalgia and irritable bowel syndrome [50]. While the efficacy of melatonin in managing acute postoperative or procedural pain remains inconclusive [50], studies on fibromyalgia have demonstrated positive outcomes with melatonin supplementation, likely due to its ability to reduce oxidative stress and enhance sleep quality [48,51]. Clinical trials have indicated that melatonin can alleviate symptoms such as thermal hyperalgesia and cold allodynia in neuropathic pain models. However, the results are mixed, as some studies show no significant analgesic effects in certain experimental contexts [48,52]. Recent research has also emphasized the role of the anterior cingulate cortex (ACC) and microglial hyperactivity in the modulation of neuropathic pain by melatonin [49].

Combining melatonin with other treatments has shown promising results. For instance, a protocol that combines pregabalin and melatonin suggests greater efficacy than monotherapy for fibromyalgia, owing to their independent mechanisms of action [51]. While preclinical studies frequently indicate robust analgesic effects, clinical trials sometimes yield inconsistent results. Factors such as dosage, route of administration, and patient variability influence outcomes [50,52]. Problems with absorption limit the effectiveness of over-the-counter melatonin supplements for pain management. Melatonin shows promise as an adjunct therapy for chronic pain conditions such as fibromyalgia and neuropathic pain [48]. However, further research is required to better understand its mechanisms and optimize its clinical application.

#### 2.3.6. Anti-Cancer Effects

Melatonin exhibits significant anti-cancer effects through various mechanisms. It inhibits cyclin-dependent kinases (CDKs) and downregulates cyclins, inducing cell cycle arrest in cancer cells. For instance, melatonin reduces CDK2/4 expression in ovarian and osteosarcoma cells and decreases levels of cyclin A/B/D/E in colon, breast, and pancreatic cancers [53]. Additionally, it promotes the expression of tumor suppressor proteins such as p21 and p27, which contribute to G2/M phase arrest in liver and breast cancer cells [53].

Melatonin selectively induces apoptosis in cancer cells by activating both intrinsic and extrinsic apoptotic pathways. It increases pro-apoptotic proteins like Bax and caspases, while decreasing levels of anti-apoptotic proteins such as Bcl-2 [53,54]. Melatonin modulates the p53 pathway, enhancing tumor suppression through mechanisms such as phosphorylation and protein stabilization [53]. Furthermore, melatonin inhibits angiogenesis by downregulating VEGF and hypoxia-inducible factor-1α (HIF-1α), critical components of the tumor blood supply system. These effects are documented in breast cancer and other malignancies [54,55].

Melatonin also reduces oxidative stress and protects cells from DNA damage induced by ROS. This antioxidant capability is essential for preventing cancer progression [54,56]. Additionally, melatonin influences oncogene expression and tumor suppressor activity by altering DNA methylation patterns and regulating microRNAs (miRNAs) [55,57]. It further diminishes cancer cell migration and invasiveness by affecting epithelial–mesenchymal transition (EMT) markers and matrix metalloproteinases (MMPs) [55,56]. Furthermore, melatonin boosts the efficacy of chemotherapy and radiotherapy, simultaneously lessening their adverse effects, including neurotoxicity, cardiotoxicity, and fatigue [53,55]. Clinical studies have demonstrated enhanced tumor regression rates when melatonin is administered in conjunction with standard treatments for cancers such as those of the lung, breast, and gastrointestinal tract [53].

Melatonin’s therapeutic potential is evident across various cancers. It inhibits proliferation through the COX-2/PGE2 pathway and induces apoptosis via caspase activation in breast cancer [54,55]. Melatonin also facilitates p21-mediated cell cycle arrest and reduces oxidative stress in liver cancer [53]. In colon cancer, melatonin decreases cyclin levels, thereby inhibiting cell division [53]. It downregulates CDK4/6 to suppress growth in osteosarcoma [53], and curbs proliferation triggered by epidermal growth factor (EGF) in prostate cancer [54]. Melatonin is promising as an adjuvant therapy due to its minimal toxicity, enhanced treatment outcomes, and its ability to ameliorate side effects, thus improving patient quality of life [53,55]. Nevertheless, more clinical trials are needed to determine the optimal dosing regimens and confirm its effectiveness across a range of cancer types.

#### 2.3.7. Collagen Synthesis

Collagen is a structural protein; it is essential for numerous bodily functions and plays a critical role in wound healing. As the scaffold for new tissue, collagen is crucial for closing wounds and providing the structure necessary for tissue regeneration. Melatonin has been extensively researched for its role in collagen synthesis, especially in wound healing, skin health, and bone metabolism. It significantly enhances collagen synthesis, thereby accelerating wound healing. This effect is due to its antioxidant properties and ability to modulate oxidative stress. Research shows that melatonin facilitates collagen deposition at wound sites, thus promoting tissue regeneration [58]. Additionally, Melatonin’s interaction with calmodulin and retinoid nuclear receptors may contribute to its regulatory effects on collagen synthesis. However, additional research is required to clarify the mechanisms involved [59].

The topical application of melatonin has been shown to enhance skin elasticity and reduce wrinkles, likely by boosting collagen production. Clinical studies using 0.1% melatonin formulations have demonstrated improvements in skin tone and micro-irritation [60]. Melatonin protects against UV-induced damage by upregulating antioxidant enzymes and mitigating DNA damage, thus indirectly supporting collagen integrity in the skin [60,61]. In vitro studies on human bone cells have shown that melatonin at concentrations of 50–100 µM significantly increases type I collagen synthesis by up to 983%, suggesting a role for melatonin in bone formation and repair [62]. Research on human cardiac fibroblasts has revealed that melatonin enhances both intracellular and extracellular collagen content at specific concentrations (e.g., 0.1 µM), though the effects vary with dosage [56].

Melatonin’s ability to neutralize free radicals is central to its role in promoting collagen synthesis, as oxidative stress can degrade collagen fibers [58,60,61]. While some fibroblasts lack membrane melatonin receptors, alternative pathways like calmodulin modulation or interactions with retinoid nuclear receptors may mediate the effects of melatonin on collagen production [59,62].

#### 2.3.8. Wound Healing

Melatonin-pretreated mesenchymal stem cell-derived exosomes have shown superior healing effects in diabetic wounds by modulating macrophage activity and reducing chronic inflammation [63]. Melatonin has been incorporated into nanogels and wound patches to enhance its delivery to wound sites. These formulations improve redox balance, protect dermal fibroblasts from UV damage, and stimulate keratinocyte activity [64,65]. Studies have shown melatonin’s potential in treating complex wounds such as burns by accelerating closure and reducing scar formation through enhanced collagen synthesis [58,64].

#### 2.3.9. Alleviation of Neuropathic Pain

Melatonin reduces oxidative stress and inflammation, key contributors to neuropathic pain. Studies indicate that melatonin alleviates thermal hyperalgesia and cold allodynia in animal models by modulating nitric oxide pathways and reversing nociceptive thresholds in spinal nerve ligation models [48,49,66]. Melatonin acts via MT2 receptors in the ACC, reducing pyramidal cell excitability and promoting M2 polarization of microglia, thereby inhibiting the release of inflammatory cytokines and mitigating symptoms of neuropathic pain [49]. Research using chronic constrictive injury rat models has demonstrated melatonin’s ability to alleviate anxiety-like and depressive behaviors associated with neuropathic pain. Behavioral tests have confirmed its anti-inflammatory and anti-apoptotic effects [66].

Administration of melatonin via intraperitoneal injection or directly to the ACC in mice has shown significant dose-dependent reductions in neuropathic pain symptoms, underscoring its analgesic properties mediated by MT2 receptors [49]. In cases of chemotherapy-induced neuropathy, melatonin reverses paclitaxel-triggered mitochondrial dysfunction by upregulating SIRT1, crucial for energy metabolism in dorsal root ganglion neurons [67]. It modulates pNEK2-dependent epigenetic pathways, normalizing TRPV1 expression and altering histone methylation (H3K27me3) to reduce neuronal hypersensitivity [68]. Co-administration of melatonin with morphine diminishes opioid tolerance by revitalizing Nrf2/HO-1 antioxidant pathways and reducing spinal neuroinflammation. Specific MT2 agonists, like IIK7, delay and reverse morphine tolerance in neuropathic rats [69,70]. Additionally, melatonin supplementation has demonstrated potential in alleviating abdominal pain related to irritable bowel syndrome and reducing symptoms of fibromyalgia, suggesting its broader utility in managing chronic pain conditions [48].

Although melatonin exhibits strong analgesic potential, the exact mechanisms by which it operates are not fully understood. Ongoing clinical trials are expected to provide detailed evidence to better define its role in managing neuropathic pain. This expanding body of research underscores melatonin’s diverse benefits for neuropathic pain relief, including anti-inflammatory effects, modulation of neuroimmune pathways, and enhancement of sleep quality.

## 3. Mechanisms of Action and Effects of PDRN

### 3.1. PDRN: A DNA-Derived Biologic Agent or Biopolymer

PDRN is a bioactive compound derived from DNA fragments (200–800 bp), primarily from salmonid gonads, utilized in wound healing, tissue regeneration, and anti-inflammatory treatments. Recent studies have broadened our understanding of its mechanisms and therapeutic potential, including innovative extraction methods and therapeutic benefits. Traditionally, PDRN is sourced from salmon sperm or human placenta with molecular weights ranging from 50 to 1500 kDa and is characterized by a high purity level (>95%) with established safety profiles in clinical settings [71,72]. A landmark study in 2025 introduced a microbial-derived PDRN (L-PDRN) from Lactobacillus rhamnosus, which showed enhanced antioxidant activity, bioavailability, and skin absorption compared to its salmon-derived counterparts. L-PDRN activates the focal adhesion kinase (FAK) and protein kinase B (AKT) signaling pathways while also engaging alternative pathways for p38 and ERK phosphorylation [73].

PDRN primarily operates by activating adenosine A2A receptors. It stimulates VEGF synthesis, which enhances angiogenesis and promotes wound healing [71,72]. PDRN reduces inflammatory cytokines such as IL-1β, IL-6, TNF-α, and iNOS, lowers apoptosis rates, and supports tissue regeneration [71,74]. It utilizes the salvage pathway to regenerate nucleotides for DNA synthesis, thereby reactivating normal cell proliferation [71].

PDRN facilitates re-epithelialization at skin graft donor sites without adverse effects [75]. In diabetic wound models, PDRN promotes angiogenesis and improves healing outcomes [76]. Combined with collagen matrices, PDRN enhances tissue repair in preclinical studies [77]. It supports tendon-to-bone healing in rotator cuff injuries and reduces fat degeneration, with clinical reports noting improvements in pain relief and functional recovery [78]. PDRN also mitigates IL-1β-induced apoptosis in human bone marrow-derived mesenchymal stem cells, fostering chondrogenic differentiation through the cAMP/PKA/CREB signaling pathway and downregulating NF-kB activation to decrease inflammation [74]. Emerging research also highlights PDRN’s anti-aging properties due to its ability to boost collagen synthesis and reduce oxidative stress, suggesting broader applications in skincare products [72].

L-PDRN presents several advantages when compared to traditional sources: (1) it demonstrates enhanced biological activity in conditions of oxidative stress; (2) it features greater scalability and environmental sustainability; (3) it exerts superior immunomodulatory effects under non-inflammatory conditions [73]. Recent studies in PDRN research have broadened its therapeutic applications in regenerative medicine, dermatology, and musculoskeletal disorders. The introduction of microbial-derived PDRN offers a promising alternative to conventional sources with improved bioactivity and sustainability.

### 3.2. Mechanisms of Action and Effects of PDRN

#### 3.2.1. Adenosine A2A Receptor Activation

PDRN serves as an agonist for the adenosine A2A receptor. This interaction is well-studied for its considerable therapeutic potential in various physiological and pathological conditions. PDRN’s activation of the adenosine A2A receptor modulates the cAMP-PKA pathway, reduces pro-inflammatory cytokines such as TNF-α and IL-1β, and inhibits apoptosis. These effects occur through the mitogen-activated protein kinase (MAPK) signaling pathway, which governs cellular responses to stress and inflammation [79,80,81]. PDRN treatment has reduced inflammation, enhanced short-term memory, and suppressed MAPK cascade activation in gerbils with induced cerebral ischemia. These therapeutic effects have been associated with elevated cAMP levels and CREB phosphorylation [80].

PDRN reduced neuronal damage, demyelination, and motor deficits in spinal cord injury models by activating the Wnt/β-catenin signaling pathway and reducing apoptosis [81]. It also alleviated inflammation and voiding dysfunction in rat models by modulating the MAPK/NF-κB pathways [79]. PDRN increases VEGF levels through adenosine A2A receptor activation, thus promoting angiogenesis, tissue repair, and metabolic activity in damaged tissues [82,83]. Additionally, it enhances VEGF-A production, improves spermatogenesis, and increases endothelial nitric oxide synthase (eNOS) activity in testicular ischemia models [82,83]. Adenosine A2A receptor activation by PDRN also increases brain-derived neurotrophic factor production in cortical neurons, enhancing neuroprotection against damage induced by cerebral ischemia [80].

#### 3.2.2. Inhibition of MAPK Signaling Pathway

PDRN inhibits the MAPK signaling pathway primarily through the activation of adenosine A2A receptors, which leads to downstream modulation of inflammatory and apoptotic processes. Recent research highlights its role in suppressing MAPK phosphorylation and the associated production of pro-inflammatory cytokines across various disease models [74,80,84]. PDRN binds to adenosine A2A receptors, raising intracellular cAMP levels. This activates the cAMP-PKA-CREB pathway, inhibiting the phosphorylation of MAPK cascade components (ERK, JNK, and p38) [74,80,84]. In CCl_4_-induced acute liver injury, PDRN reduced the phosphorylation of IκB-α, ERK, JNK, and p38, thereby inactivating the NF-κB and MAPK pathways [84]. In cerebral ischemia models, PDRN inhibited MAPK phosphorylation (ERK, JNK, and p38) by increasing cAMP, which mitigated neuroinflammation and apoptosis [80]. It also decreased levels of TNF-α, IL-1β, and IL-6 in liver and brain tissues by blocking NF-κB/MAPK activation [80,84].

In osteoarthritis, PDRN reversed IL-1β-induced chondrocyte damage by restoring the cAMP/PKA/CREB signaling, indirectly suppressing MAPK-driven inflammation [74]. Co-treatment with DMPX (3,7-dimethyl-1-propargylxanthine), an adenosine A2A receptor antagonist, negated PDRN’s inhibitory effects on MAPK phosphorylation and cytokine production, confirming receptor-dependent action [74,80,84]. Studies have demonstrated PDRN’s efficacy in conditions like acute liver injury [81], cerebral ischemia [80], and degenerative joint diseases [74], positioning it as a multifaceted anti-inflammatory agent. Its ability to modulate MAPK signaling through adenosine A2A receptors presents a promising route for treating inflammatory and oxidative stress-related conditions. Research is ongoing to optimize dosing and delivery methods to enhance its clinical application [74,84].

#### 3.2.3. Enhancement of Collagen Production

PDRN has shown significant potential in enhancing collagen production through various mechanisms, as supported by recent studies. It stimulates adenosine A2A receptors, triggering downstream pathways that reduce pro-inflammatory cytokines (TNF-α, IL-6) while elevating anti-inflammatory IL-10 [85,86]. This anti-inflammatory environment supports collagen synthesis by fibroblasts [87]. PDRN increases VEGF-A output by 42%, which promotes angiogenesis and nutrient transport to collagen-producing cells [85,88].

An enhanced blood supply bolsters fibroblast activity and extracellular matrix (ECM) remodeling [84]. PDRN also activates the FAK-AKT pathway, vital for cell migration and tissue repair, while regulating ECM synthesis through phosphorylation of p38 and ERK [73]. L-PDRN amplifies these effects with its smaller DNA fragments (<100 bp), improving skin absorption and bioavailability [73].

L-PDRN outperforms salmon-derived PDRN in enhancing collagen function. Its smaller fragment size leads to deeper skin penetration, resulting in quicker wound healing and enhanced ECM regeneration under inflammatory conditions [73]. Collagen matrices loaded with 2 mg/mL PDRN achieved keratinized tissue heights comparable to those of free gingival grafts in canine models, showcasing its effectiveness in oral soft tissue regeneration [77]. When combined with collagenated biphasic calcium phosphate, PDRN enhanced early bone formation in lateral sinus augmentations, linking angiogenesis to the development of collagen-rich osteoid tissue [88]. The degradation of PDRN releases nucleotides that fuel the production of collagen types I and III, elastin, and fibrinogen by fibroblasts via the salvage pathway, thereby reducing the energy demands for ECM synthesis [87].

Recent advancements have highlighted microbial-derived PDRN as a sustainable alternative with enhanced bioactivity [73]. However, further research is required to optimize dosing in collagen matrices [77] and to validate long-term ECM stability in human trials [87]. In summary, the collagen-enhancing effects of PDRN are driven by anti-inflammatory modulation, growth factor induction, and targeted signaling pathways, with microbial-derived variants offering promising improvements in scalability and efficacy.

#### 3.2.4. Promotion of Angiogenesis

PDRN demonstrates significant pro-angiogenic properties through multiple molecular mechanisms and therapeutic applications. Recent studies emphasize its role in promoting wound healing, tissue regeneration, and disease management through angiogenesis modulation. PDRN activates the adenosine A2A receptor, stimulating production of VEGF and expression of CD31 to promote blood vessel formation [89]. This pathway also enhances pro-angiogenic factors such as platelet-derived growth factor and angiopoietin-2, and inhibits anti-angiogenic factors, including endostatin and angiostatin [90]. Additionally, PDRN modulates the Janus Kinase (JAK)/Signal Transducer and Activator of Transcription (STAT) signaling pathway, reducing the phosphorylation of JAK1, JAK2, STAT1, and STAT3, which aids in mitigating inflammation and supporting vascular repair [91].

In osteoarthritis chondrocyte models, PDRN was shown to increase VEGF levels by 41.6% and improve wound closure rates by 41.62% compared to controls, while also reducing the expression of the catabolic enzyme MMP-13 [90]. Studies on diabetic fibroblasts revealed that PDRN-loaded alginate hydrogels significantly enhance the expression of fibroblast growth factor and VEGF by 2.5-fold, thereby accelerating cell migration and angiogenesis [92]. In diabetic mouse models, PDRN hydrogels were found to improve wound healing by increasing collagen density and VEGF expression. PDRN hydrogels facilitated faster wound healing in diabetic mice by boosting collagen (55% compared to 30% in the control group) and VEGF (56% compared to 30%) levels in the wounds [89]. Burn injury models have shown that PDRN-induced re-epithelialization and angiogenesis occur via upregulation of VEGF/CD31 [89].

Sustained delivery of PDRN through hydrogels has proven more effective than single injections in promoting vessel density and reducing inflammatory markers such as TGF-β and myeloperoxidase [92]. Moreover, the angiogenic effects of PDRN were negated by adenosine A2A antagonists, confirming the receptor-dependent mechanisms involved [89].

The dual action of PDRN in promoting pro-angiogenic factors while inhibiting catabolic pathways positions it as a potent regenerative therapy for several applications: (1) enhanced vascularization and collagen synthesis in diabetic wounds [86,89]; (2) cartilage repair through anabolic stimulation in osteoarthritis [90]; (3) accelerated re-epithelialization and restoration of blood flow in thermal injuries [92].

However, limitations include reliance on cell-line models for osteoarthritis studies [90] and the need for chronic disease models to validate long-term efficacy [90].

#### 3.2.5. Stimulation of Tissue Repair

PDRN has demonstrated significant potential in tissue repair through various mechanisms, including activation of salvage pathways, modulation of adenosine receptors, and induction of growth factors. It has shown efficacy in wound healing, diabetic complications, and regenerative medicine, enhanced by innovations in microbial-derived formulations.

PDRN promotes tissue regeneration by supplying nucleotides via the salvage pathway, which bypasses impaired de novo DNA synthesis in hypoxic or damaged tissues, thereby reactivating cell proliferation [93,94]. Furthermore, PDRN’s binding to adenosine A2A receptors triggers downstream pathways that enhance angiogenesis and reduce inflammation [88,90]. For instance, activation of adenosine A2A increases VEGF expression, crucial for restoring blood flow in ischemic conditions [76,93].

In diabetic models, PDRN accelerates wound closure by upregulating VEGF, CD31, and collagen synthesis, counteracting impaired angiogenesis [76,93]. Studies in 2019 demonstrated that PDRN reduced wound depth by 40% in diabetic mice through enhanced dermal regeneration [76]. Additionally, its anti-inflammatory properties aid healing by suppressing TNF-α and metalloproteinases (MMP-2/9), which degrade the extracellular matrix [73,93]. Clinical trials have reported reduced healing time and minimal side effects, supporting its utilization in chronic wounds [94].

PDRN enhances outcomes in diabetes-related complications, such as impaired tendon repair. A 2024 study found that PDRN administration in diabetic rats increased the rotator cuff’s load-to-failure by 35% and decreased fatty infiltration by stimulating VEGF and FGF [95]. For peripheral artery disease, PDRN augmented femoral artery blood flow by 50% via VEGF-driven angiogenesis, an effect blocked by A2A antagonists [91,93].

A groundbreaking 2025 study introduced L-PDRN, showing superior bioavailability and antioxidant capacity compared to traditional salmon-derived PDRN [73]. Containing smaller DNA fragments (<100 bp), L-PDRN enhances skin absorption and activates dual pathways (FAK-AKT and p38/ERK) to accelerate wound healing under inflammatory conditions [73]. It also exhibits immunomodulatory potential, increasing nitric oxide production in macrophages while reducing lipopolysaccharide-induced inflammation [73].

Unlike defibrotide, an anti-angiogenic DNA drug, PDRN uniquely combines pro-angiogenic and anti-inflammatory effects [93]. Ongoing studies emphasize the importance of rigorous bioequivalence testing for new PDRN formulations, as molecular weight variations significantly impact therapeutic outcomes [73,93]. No adverse effects have been reported in clinical applications to date [94,95].

#### 3.2.6. Melanin Reduction

PDRN inhibits melanogenesis by suppressing microphthalmia-associated transcription factors and downstream targets like tyrosinase and tyrosinase-related protein-1, achieved via activation of the ERK and AKT pathways, which downregulate MITF expression [96]. Additionally, combined with niacinamide and vitamin C, PDRN reduces mitochondrial oxidative stress by enhancing nicotinamide nucleotide transhydrogenase expression, further decreasing melanogenic signals such as MC1R, TYRP1, and TYRP2 [97].

In murine melanocytes (Mel-Ab cells), PDRN reduced melanin content by 25–40% and tyrosinase activity by 30–50% in a dose-dependent manner [96]. In human melanocyte–keratinocyte cocultures, PDRN treatment resulted in a 50% reduction in melanin content, confirming its efficacy within a skin-like microenvironment [96]. Additionally, L-PDRN from Lactobacillus rhamnosus has shown superior antioxidant activity compared to salmon-derived PDRN, potentially enhancing its anti-melanogenic effects through improved bioavailability [73].

In a study involving six patients with facial hyperpigmentation, three sessions of intradermal PDRN injections resulted in ≥50% improvement in pigmented lesions [96]. Post-thyroidectomy patients receiving PDRN injections experienced a 38% reduction in their melanin index and a decrease in scar erythema, compared to controls [98].

The combination of PDRN with niacinamide and vitamin C effectively reduced melanosome transfer by downregulating RAB27A and RAB32 proteins, offering a multi-target approach to hyperpigmentation [97]. Introduced in 2025, L-PDRN represents a sustainable alternative with enhanced bioactivity for future dermatological applications [73].

These findings establish PDRN as a versatile agent for managing hyperpigmentation, with ongoing innovations in formulation and delivery expanding its therapeutic potential.

## 4. Comparison of the Effects of Melatonin and PDRN

Table 1 summarizes the various effects of melatonin and PDRN, highlighting both similarities and distinctions between the two compounds.

Melatonin is recognized for its antioxidant activity, anti-inflammatory effects, ability to regenerate nerve cells, and protective functions for brain health, with potential anti-cancer effects [23,32,33,34,35,36,37,38,39,59,60,61,62,63,99,100,101,102,103,104,105,106,107,108,109,110,111,112,113]. PDRN also exhibits significant anti-inflammatory properties and promotes cell and tissue regeneration, tissue repair, and angiogenesis, the process of new blood vessel formation [73,74,76,79,89,90,91,92,93,94,95,108,109,110,111,112,113,114,115,116,117,118,119,120,121,122,123].

Both melatonin and PDRN have significant anti-inflammatory properties, underscoring their potential roles in regenerative and protective functions at cellular and tissue levels [23,32,33,34,35,36,37,38,39,53,54,55,56,57,73,74,76,79,89,90,91,92,93,94,95,99,100,101,102,103,104,105,106,107,108,109,110,111,112,113,114,115,116,117,118,119,120,121,122,123,124,125,126,127,128,129]. They also contribute to healing processes in various tissues, suggesting potential applications in regenerative medicine. Additionally, both compounds may have beneficial effects in the treatment of cancer, albeit possibly through different mechanisms.

Regarding their mechanisms of action, melatonin primarily functions by acting on MT1 and MT2 receptors, scavenging free radicals, and facilitating the removal of waste products through the cerebrospinal fluid [12,23,24,25,26,27,28,29,30,31,32,33,34,35,36,124,125,126,130,131,132,133,134,135,136,137]. Conversely, PDRN activates adenosine A2A receptors and stimulates VEGF synthesis, which promotes angiogenesis [5,6,79,80,81,82,83,91,93,117,120,123,138,139,140].

In their effects, melatonin is known to prevent cellular aging and enhance brain function while inhibiting the growth of cancer cells [48,49,50,51,52,53,54,55,56,57,58,59,60,61,62,63,64,65,102,103,104,105,106,107,128,141,142,143,144,145,146,147,148,149,150,151,152,153]. PDRN, on the other hand, promotes the growth of damaged tissue and increases the secretion of fibroblasts and growth factors, thereby facilitating tissue regeneration across a broad range of conditions [5,73,77,85,86,87,88,90,95,108,111,113,114,115,116,117,118,121,123,154,155,156,157,158,159,160,161].

Their applications also highlight both similarities and distinctions. Both are employed in skin regeneration and esthetic enhancement; however, PDRN specifically targets the regeneration of ligaments, muscles, tendons, and cartilage, and is used in the healing of ulcers and treatment of musculoskeletal disorders [5,73,93,95,96,97,113,117,122,138,162,163,164,165,166,167,168,169,170,171,172,173,174,175,176,177,178]. Melatonin, associated with skin and brain health, is additionally utilized in treating sleep disorders and improving cognitive function [53,54,55,56,57,65,102,103,104,105,106,107,144,145,146,147,179,180,181,182,183,184,185,186,187,188].

Overall, melatonin and PDRN target complementary therapeutic areas, with shared benefits in cellular health and potential for hospital synergy, yet they differ notably in their mechanisms and specific applications across various medical fields.

**Table 1 ijms-26-05703-t001:** Comparison of the effects of melatonin and PDRN.

	Melatonin [References ^1^]	PDRN [References ^1^]
Main roles	- Antioxidant activity [32,33,34,35,36,124,125,126] - Anti-inflammatory effects [23,37,38,39,124,127,128] - Regeneration of nerve cells [99,100,101,129] - Protection of brain function [102,103] - Anti-cancer effects [53,54,55,56,57,104,105,106,107]	- Anti-inflammatory effects [79,108,109,110,111,112] - Promotion of cell and tissue regeneration [111,113,114,115,116,117] - Tissue repair [73,74,76,91,93,94,95,113,115,118,119,120] - Promotion of angiogenesis [74,89,90,91,92,114,121,122,123]
Mechanisms of action	- Acting via MT1 and MT2 receptors [12,23,24,25,26,27,28,29] - Scavenging of free radicals [32,33,34,35,36,124,125,126,130,131,132,133,134] - Waste elimination via cerebrospinal fluid [132,133,134]	- Stimulation of adenosine A2A receptors [79,80,81,82,83,91,93,117,120,138,139] - Induction of VEGF synthesis [5,6,123,140]
Main effects	- Prevention of cellular aging [141,142,143] - Enhancement of brain function [102,103] - Inhibition of cancer cell growth [53,54,55,56,57,104,105,106,107,141,142,143,144] - Boosting of collagen synthesis and wound healing [58,59,60,61,62,63,64,65,148] - Alleviation of neuropathic pain [48,49,50,51,52,128,149,150,151,152,153]	- Stimulation of cell growth in damaged tissue [111,113,114,115,116,117] - Enhanced secretion of fibroblasts and growth factors [95,123,154,155,158] - Promoting wound healing [113,115,116,118,121,123,157] - Relieving pain [90,158,159] - Stimulating collagen synthesis [5,73,77,85,86,87,88,108,118,157,160,161]
Applications	- Enhancing skin regeneration and beauty [65,179,180,181,182] - Enhancing and protecting brain function [102,103] - Assisting in cancer treatment [53,54,55,56,57,104,105,106,107,144,145,146,147] - Alleviating sleep disorders [183,184,185,186,187,188]	- Skin regeneration and beauty [73,96,97,113,117,122,162,163,164,165,166] - Regenerating damaged ligaments, muscles, tendons, and cartilage [95,167,168,169,170,171,172,173] - Healing ulcers [5,93,138,174] - Treating musculoskeletal disorders such as plantar fasciitis and lateral epicondylitis [175,176,177,178]
Main sources of supply	- Pineal gland in the brain (produced by the body) - Dietary supplements (synthetic melatonin)	- Fish sperm or semen (particularly from salmon) - Plants (e.g., mugwort and broccoli)

^1^ All references related to the table above have been comprehensively organized based on the most recent research articles and review papers from 2020 to 2025.

## 5. Comprehensive Research Framework

### 5.1. Potential Synergistic Effects

Most recently, the use of a bioactive scaffold of mesenchymal progenitor cells in combination with PDRN and melatonin to improve ovarian function was announced, with the potential to offer therapeutic benefits [189]. Therefore, the theoretical combination of melatonin and PDRN could potentially offer enhanced therapeutic applications and benefits, presenting a novel concept with promising implications:Enhanced Anti-inflammatory Response: Melatonin is known to suppress pro-inflammatory cytokines, while PDRN activates adenosine A2A receptors to reduce inflammation. The combined action of both compounds could potentially result in a more potent anti-inflammatory effect.Augmented Anti-Inflammatory Effects: Both melatonin and PDRN independently exhibit anti-inflammatory properties. Utilizing them together could potentially provide a more comprehensive reduction in inflammation, which would be beneficial in conditions characterized by excessive inflammatory responses.Comprehensive Inflammatory Pathway Modulation: Melatonin acts by inhibiting inflammasome activity, whereas PDRN suppresses MAPK signaling. Together, they can address inflammation at multiple regulatory points, potentially leading to a more effective overall anti-inflammatory response.Enhanced Antioxidant Defense: Melatonin’s ability to boost antioxidant enzyme activity combined with PDRN’s role in promoting tissue repair through VEGF induction could synergize to reduce oxidative stress more effectively and facilitate healing processes.Accelerated Tissue Repair: Melatonin’s antioxidant properties protect tissues from oxidative damage, while PDRN promotes tissue regeneration. Together, these compounds may enhance healing processes in conditions such as skin injuries or degenerative diseases.Enhanced Tissue Regeneration: Melatonin’s antioxidant properties, coupled with PDRN’s promotion of angiogenesis and cell proliferation, could synergistically accelerate tissue repair processes.Improved Pain Management: Both melatonin and PDRN possess analgesic properties: melatonin modulates pain perception, while PDRN addresses inflammation-induced pain. Utilizing them in combination could potentially provide comprehensive pain relief.Potential in Esthetic Medicine: Given PDRN’s capacity to reduce melanin synthesis and melatonin’s regulatory effects on skin cells, their combination might offer innovative solutions in cosmetic treatments aimed at skin rejuvenation and mitigating pigmentation disorders.

### 5.2. Potential Clinical Applications

Several clinical scenarios could benefit from the combined use of melatonin and PDRN:Chronic Pain Conditions: The integration of melatonin’s analgesic effects with PDRN’s anti-inflammatory properties may offer effective relief in chronic pain syndromes.Degenerative Joint Diseases: In conditions like osteoarthritis, where inflammation and tissue degeneration prevail, this combination could reduce inflammation and promote cartilage repair.Esthetic Medicine: For skin rejuvenation, the antioxidant effect of melatonin combined with PDRN’s collagen-stimulating properties may enhance skin elasticity and reduce signs of aging.

### 5.3. Future Directions

While the individual benefits of melatonin and PDRN are well-documented, research exploring their combined use is limited. Preliminary data suggest that such combinations could enhance therapeutic outcomes, particularly in regenerative medicine and dermatology. Nonetheless, comprehensive clinical trials are essential to establish optimal dosing regimens, safety profiles, and the full spectrum of therapeutic benefits. Although preclinical studies have indicated potential synergistic effects, robust clinical trials are required to validate these findings. Future research should focus on the following areas:Dose Optimization: Determining the optimal dosages for combination therapy to maximize benefits and minimize potential side effects.Preclinical Studies: Utilizing animal models to assess the combined effects of melatonin and PDRN on tissue repair, inflammation, and markers of oxidative stress.Mechanistic Studies: Investigating the molecular and signaling pathways, such as SIRT1, Nrf2, and NF-κB, involved in their combined action to better understand the synergistic effects.Clinical Trials: Conducting randomized controlled trials to evaluate the safety and efficacy of combined melatonin and PDRN therapy in various patient populations, focusing on conditions such as neurodegenerative diseases, ischemic injuries, and dermatological conditions to comprehensively assess efficacy and safety.Delivery Systems: Developing advanced delivery mechanisms, such as nanocarriers, to enhance the bioavailability and targeted delivery of melatonin and PDRN, thereby maximizing therapeutic outcomes.

## 6. Conclusions

This paper highlights the promising therapeutic potential of combining melatonin and PDRN, leveraging their distinct pharmacological properties. Melatonin, known for modulating sleep, oxidative stress, and inflammation, complements PDRN’s ability to promote tissue regeneration and angiogenesis. Individually, they offer significant benefits; however, their combined anti-inflammatory and antioxidant effects are proposed to profoundly enhance healing processes across various tissues, particularly in conditions marked by inflammation, oxidative stress, and tissue degeneration.

Given these analytical considerations and existing research, co-administering melatonin and PDRN may provide a synergistic approach to managing oxidative stress and tissue damage, with broad implications for wound healing and beyond (Figure 1).

While the combination shows considerable promise, extensive empirical validation through comprehensive clinical trials is paramount. These studies are not just beneficial, they are crucial for developing standardized treatment protocols that fully unlock the therapeutic potential of these agents. Future research must optimize these combination therapies to definitively confirm that their complementary mechanisms offer enhanced benefits over individual uses. Ultimately, this review underscores the urgent need for in-depth research to fully understand their synergistic effects and establish effective therapeutic strategies across diverse medical fields.

## Figures and Tables

**Figure 1 ijms-26-05703-f001:**
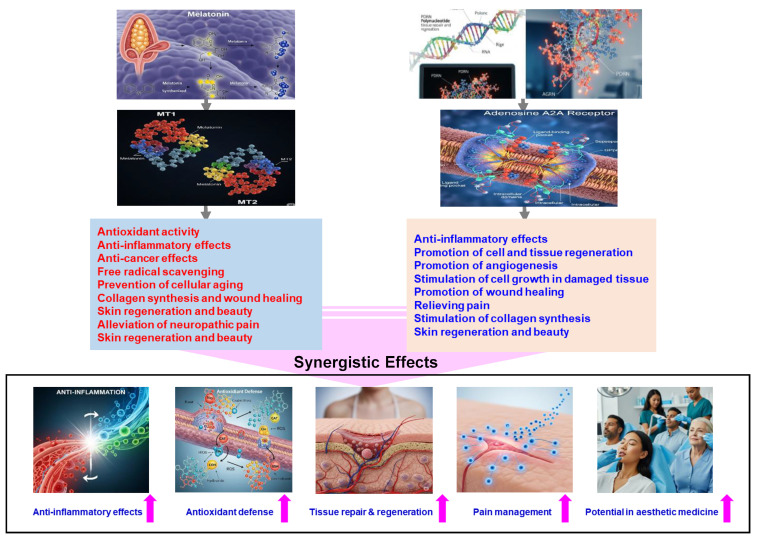
Summary of the possible synergistic effects of combining melatonin and PDRN.

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
