# Peer review of "Comparative Analysis of Melatonin and Polydeoxyribonucleotide: Possible Benefits of Co-Treatment Effects and Potential Synergistic Applicability"

_ijms, 2025, doi:10.3390/ijms26125703_

Round 1

Reviewer 1 Report

Comments and Suggestions for Authors

In present work, Jang et al. try to review the pharmacological effects and mechanisms of melatonin and polydeoxyribonucleotide (PDRN), suggesting that their integration could amplify their individual benefits, particularly in wound healing and various medical applications. However, there are some questions that should be explained.
Major concerns
1. As a review article, there is only one Figure. Figures related to the mechanisms of action and effects of melatonin and PDRN are needed.
2. Melatonin is also involved in immunoregulation via MT1 and MT2 in immune organs, which is related to their pharmacological effects. These contents should be added.
3. Figure 1 is in low quality, and it should be included some cartoon pictures.
4. A closely related article is not cited by this review paper. Combination therapy of melatonin and PDRN is used to restore ovarian function in this paper.
Kim DS, Jeong DS, Park SY, Jung JW, Lee JE, Lee JK, Baek SW, Lee DR, Han DK. Ovarian Function Restoration with Biomimetic Scaffold Incorporating Angiogenic Molecules and Antioxidant in Chemotherapy-Induced Perimenopausal Model. Adv Healthc Mater. 2025;10:e2403944.
5. English grammar and writing should be checked and revised throughout the manuscript. 
Minor concerns
1. Abstract section should be rewritten. A simple background at the beginning and conclusion at the end are needed.
2. Line 35, please explain ‘PDRN’.
3. There are many ‘we’ (Lines 41, 48, etc.). As a scientific paper, it is necessary to use the third person style. 
4. Line 95, delete ‘(DSWPD)’.
5. Lines 104-105, revise ‘These findings highlight the critical need for more stringent regulatory oversight to ensure products are accurately labeled and free from contaminants.’.
6. Line 134, delete ‘(GPCRs)’.
7. Line 362, delete ‘(MSC)’.
8. Line 377, delete ‘(CCI)’.
9. Line 385, delete ‘(DRG)’.
10. Line 443, please explain ‘MAPK’.
11. Line 456, delete ‘(BDNF)’.
12. Line 518, delete ‘(PDGF)’.
13. Line 519, delete ‘(ANG-2)’.
14. Line 520, please explain ‘JAK/STAT’.
15. ‘5. Potential Synergistic Effects’, ‘6. Clinical Applications’, and ‘7. Current Research and Future Directions’ are the authors’ speculation, which should be merged into one section. There is no a reference in ‘7. Current Research and Future Directions’ section, so ‘Current Research and’ should be deleted.
16. Conclusions section should be rewritten.

Comments on the Quality of English Language

The English could be improved to more clearly express the research.

Author Response

May 30, 2025

Dear Reviewer 1,

Thank you for your letter and for the reviewers’ comments concerning our manuscript entitled " Comparative Analysis of Melatonin and Polydeoxyribonucleotide: Treatment Effects and Potential Synergistic Applicability."

We completely agree with your suggestions regarding our manuscript. The manuscript is completely revised as you and your colleague requested.

The corrections and revisions are as follows:

In present work, Jang et al. try to review the pharmacological effects and mechanisms of melatonin and polydeoxyribonucleotide (PDRN), suggesting that their integration could amplify their individual benefits, particularly in wound healing and various medical applications. However, there are some questions that should be explained.
Major concerns
1. As a review article, there is only one Figure. Figures related to the mechanisms of action and effects of melatonin and PDRN are needed.
Answer: Thank you very much for your good comments.

Figure 1 is reproduced again.

  1. Melatonin is also involved in immunoregulation via MT1 and MT2 in immune organs, which is related to their pharmacological effects. These contents should be added.

Answer: Thank you very much for your good comments.

As the reviewer pointed out, immunomodulation is the most important action of melatonin.

However, for the purpose of this review paper, we almost omitted immunomodulation and emphasised anti-inflammatory and anti-cancer effects as anti-cancer effects and inhibition of cancer cell growth to highlight the commonality with the action of PDRN. We hope you will understand the shortcomings.

  1. Figure 1 is in low quality, and it should be included some cartoon pictures.

Answer: Thank you very much for your good comments.

Figure 1 is reproduced again.

  1. A closely related article is not cited by this review paper. Combination therapy of melatonin and PDRN is used to restore ovarian function in this paper.
    Kim DS, Jeong DS, Park SY, Jung JW, Lee JE, Lee JK, Baek SW, Lee DR, Han DK. Ovarian Function Restoration with Biomimetic Scaffold Incorporating Angiogenic Molecules and Antioxidant in Chemotherapy-Induced Perimenopausal Model. Adv Healthc Mater. 2025;10:e2403944.

Answer: Thank you very much for your good comments.

We have added the literature to the text lines 640-642 and also added it as reference 186.

Lines 640-642: Most recently, the use of a bioactive scaffold of mesenchymal progenitor cells in combination with PDRN and melatonin to improve ovarian function was announced, with the potential to offer therapeutic benefits [186]. Therefore, the theoretical combination….

  1. Kim, D.S.; Jeong, D.S.; Park, S.Y.; Jung, J.W.; Lee, J.E.; Lee, J.K.; Baek, S.W.; Lee, D.R.; Han, D.K. Ovarian Function Restoration with Biomimetic Scaffold Incorporating Angiogenic Molecules and Antioxidant in Chemotherapy-Induced Perimenopausal Model. Adv. Healthc. Mater. 2025, 14, e2403944. doi: 10.1002/adhm.202403944.
  2. English grammar and writing should be checked and revised throughout the manuscript. 

Answer: Thank you very much for your good comments.

I received English proofreading through a proofreading company (Harrisco). The Certificate of Editing is attached below.

I am very sorry if you think that the reviewer must have been under considerable stress while reviewing the thesis. However, we also feel that it is too insufficient even though we have received the maximum correction through the correction company. However, since we did our best, we are lacking, but we hope you will understand.

Minor concerns
1. Abstract section should be rewritten. A simple background at the beginning and conclusion at the end are needed.

Answer: Thank you very much for your good comments.

We've added a conclusion at the end.

Lines 25-29: In conclusion, further research is needed to optimize combination therapies involving melatonin and PDRN, with the goal of confirming their enhanced benefits when used together. This review highlights the importance of exploring their synergistic effects and developing effective therapeutic strategies across various medical fields.

  1. Line 35, please explain ‘PDRN’.

Answer: Thank you very much for your good comments.

Added as “Polydeoxyribonucleotide (PDRN)”.

  1. There are many ‘we’ (Lines 41, 48, etc.). As a scientific paper, it is necessary to use the third person style. 

Answer: Thank you very much for your good comments.

We've changed the sentence.

Lines 45-47: Therefore, the distinct mechanisms of action and varied pharmacological effects of each biomolecule are reviewed to compare and analyze the therapeutic potential of melatonin and PDRN.

Lines 51-53: By reviewing existing studies and exploring potential effects, this review aimed to provide a comprehensive understanding of the interactions between melatonin and PDRN.

  1. Line 95, delete ‘(DSWPD)’.

Answer: Thank you very much for your good comments.

We removed.

  1. Lines 104-105, revise ‘These findings highlight the critical need for more stringent regulatory oversight to ensure products are accurately labeled and free from contaminants.’.

Answer: Thank you very much for your good comments.

We removed the sentence.

  1. Line 134, delete ‘(GPCRs)’.

Answer: Thank you very much for your good comments.

We removed.

  1. Line 362, delete ‘(MSC)’.

Answer: Thank you very much for your good comments.

We removed.

  1. Line 377, delete ‘(CCI)’.

Answer: Thank you very much for your good comments.

We removed.

  1. Line 385, delete ‘(DRG)’.

Answer: Thank you very much for your good comments.

We removed.

  1. Line 443, please explain ‘MAPK’.

Answer: Thank you very much for your good comments.

We added.

  1. Line 456, delete ‘(BDNF)’.

Answer: Thank you very much for your good comments.

We removed.

  1. Line 518, delete ‘(PDGF)’.

Answer: Thank you very much for your good comments.

We removed.

  1. Line 519, delete ‘(ANG-2)’.

Answer: Thank you very much for your good comments.

We removed.

  1. Line 520, please explain ‘JAK/STAT’.

Answer: Thank you very much for your good comments.

We added.

  1. ‘5. Potential Synergistic Effects’, ‘6. Clinical Applications’, and ‘7. Current Research and Future Directions’ are the authors’ speculation, which should be merged into one section. There is no a reference in ‘7. Current Research and Future Directions’ section, so ‘Current Research and’ should be deleted.

Answer: Thank you very much for your good comments.

We replaced it with the following:

  1. Conclusion

5.1. Potential Synergistic Effects

5.2. Clinical Applications

5.3. Future Directions

5.4. Summary

  1. The Conclusions section should be rewritten.

Answer: Thank you very much for your good comments.

We rewrote it again.

Lines 717-734: This paper highlights the promising therapeutic potential of combining melatonin and PDRN, leveraging their distinct pharmacological properties. Melatonin, known for modulating sleep, oxidative stress, and inflammation, complements PDRN's ability to promote tissue regeneration and angiogenesis. Individually, they offer significant benefits; however, their combined anti-inflammatory and antioxidant effects are proposed to profoundly enhance healing processes across various tissues, particularly in conditions marked by inflammation, oxidative stress, and tissue degeneration.

Given these analytical considerations and existing research, co-administering melatonin and PDRN may provide a synergistic approach to managing oxidative stress and tissue damage, with broad implications for wound healing and beyond (Figure 1).

While the combination shows considerable promise, extensive empirical validation through comprehensive clinical trials is paramount. These studies are not just beneficial, they're crucial for developing standardized treatment protocols that fully unlock the therapeutic potential of these agents. Future research must optimize these combination therapies to definitively confirm that their complementary mechanisms offer enhanced benefits over individual uses. Ultimately, this review underscores the urgent need for in-depth research to fully understand their synergistic effects and establish effective therapeutic strategies across diverse medical fields.

Comments on the Quality of English Language

The English could be improved to more clearly express the research.

Answer: Thank you very much for your good comments.

The Certificate of Editing is attached.

**All of the edited sections and references were changed with the blue words.

***The Certificate of Editing is attached.

I hope that the revised manuscript is now acceptable for publication in the IJMS. We are looking forward to receiving your answer soon.

Sincerely,

Yeong-Min Yoo Ph.D 

Institute of Environmental Research,

Kangwon National University,

Chuncheon 24341, Republic of Korea

Email: yyeongm@hanmail.net

Reviewer 2 Report

Comments and Suggestions for Authors

Journal: IJMS (ISSN 1422-0067) Manuscript ID: ijms-3630748 Type: Review Title: Comparative Analysis of Melatonin and Polydeoxyribonucleotide: Treatment Effects and Potential Synergistic Applicability Authors: Su Kil Jang , Jaeseok Choi , Hye Won Lim , Hong-Gyum Kim , Yeong-Min Yoo * Section: Molecular Endocrinology and Metabolism Special Issue: A Moving Frontline in the Study of Melatonin and Its Analogs The review entitled “Comparative Analysis of Melatonin and Polydeoxyribonucleotide: Treatment Effects and Potential Synergistic Applicability” is interesting and in-depth in its aspects. This review examines the distinct mechanisms of action and the varied pharmacological effects of each biomolecule to compare and analyse the different aspects of melatonin (also highlighting its antioxidant aspect) and polydeoxyribonucleotide, focusing on their therapeutic potential. The authors also emphasise the synergistic effects observed when these two substances are combined. The review is topical and of great application, and this study could have a significant impact because it could provide novel therapeutic approaches and ultimately lay the groundwork for developing new and effective treatments across various applications. I think the review can be accepted for publication in the IJMS journal, after minor revisions. • Introduction section, Lines 28-49: The authors are missing the time range of the review of published articles (last 5 years, last 10 years...?): add it. • Introduction section, Lines 28-49: Information on the bibliographic search performed is missing. Where was the search performed? (…..PubMed, Scopus, Web of Science, the World Wide Web). It is important for the readers. What words are used as keywords during the literature search? • At the end of the manuscript, I would suggest adding a subsection (under Author Contributions and Funds) with the abbreviations used in the text that would help the reader. (MDPI normally provides them, and they are very useful) • Table 1: The authors reported: “1All references related to the table above have been comprehensively organised based on the most recent research literature and review papers”, but the time range is missing in all text. Specify this aspect also in this sentence.

Author Response

May 30, 2025

Dear Reviewer 2,

Thank you for your letter and for the reviewers’ comments concerning our manuscript entitled " Comparative Analysis of Melatonin and Polydeoxyribonucleotide: Treatment Effects and Potential Synergistic Applicability."

We completely agree with your suggestions regarding our manuscript. The manuscript is completely revised as you and your colleague requested.

The corrections and revisions are as follows:

Journal: IJMS (ISSN 1422-0067) Manuscript ID: ijms-3630748 Type: Review Title: Comparative Analysis of Melatonin and Polydeoxyribonucleotide: Treatment Effects and Potential Synergistic Applicability Authors: Su Kil Jang , Jaeseok Choi , Hye Won Lim , Hong-Gyum Kim , Yeong-Min Yoo * Section: Molecular Endocrinology and Metabolism Special Issue: A Moving Frontline in the Study of Melatonin and Its Analogs The review entitled “Comparative Analysis of Melatonin and Polydeoxyribonucleotide: Treatment Effects and Potential Synergistic Applicability” is interesting and in-depth in its aspects. This review examines the distinct mechanisms of action and the varied pharmacological effects of each biomolecule to compare and analyse the different aspects of melatonin (also highlighting its antioxidant aspect) and polydeoxyribonucleotide, focusing on their therapeutic potential. The authors also emphasise the synergistic effects observed when these two substances are combined. The review is topical and of great application, and this study could have a significant impact because it could provide novel therapeutic approaches and ultimately lay the groundwork for developing new and effective treatments across various applications. I think the review can be accepted for publication in the IJMS journal, after minor revisions.

1. Introduction section, Lines 28-49: The authors are missing the time range of the review of published articles (last 5 years, last 10 years...?): add it.
Answer: Thank you very much for your good comments.

We have added the literature to the text lines 53-57.

Lines 53-57: The papers utilized in this review were predominantly recent publications from 2020 to 2025, selected through a comprehensive PubMed search using 'melatonin' and 'PDRN' as primary keywords, along with additional terms such as 'receptor-mediated actions,' 'antioxidant activity,' and 'anti-inflammatory effects,' among others.

  1. Introduction section, Lines 28-49: Information on the bibliographic search performed is missing. Where was the search performed? (…..PubMed, Scopus, Web of Science, the World Wide Web). It is important for the readers. What words are used as keywords during the literature search?

Answer: Thank you very much for your good comments.

As the reviewer pointed out, immunomodulation is the most important action of melatonin.

We have added the literature to the text lines 53-57.

Lines 53-57: The papers utilized in this review were predominantly recent publications from 2020 to 2025, selected through a comprehensive PubMed search using 'melatonin' and 'PDRN' as primary keywords, along with additional terms such as 'receptor-mediated actions,' 'antioxidant activity,' and 'anti-inflammatory effects,' among others.

  1. At the end of the manuscript, I would suggest adding a subsection (under Author Contributions and Funds) with the abbreviations used in the text that would help the reader. (MDPI normally provides them, and they are very useful).

Answer: Thank you very much for your good comments.

We have added the Abbreviation section to lines 752-788.

  1. Table 1: The authors reported: “1All references related to the table above have been comprehensively organised based on the most recent research literature and review papers”, but the time range is missing in all text. Specify this aspect also in this sentence.

Answer: Thank you very much for your good comments.

We rewrote the sentence to show the year again.

Lines 646-647: All references related to the table above have been comprehensively organized based on the most recent research articles and review papers from 2020 to 2025.

**All of the edited sections and references were changed with the blue words.

***The Certificate of Editing is attached.

I hope that the revised manuscript is now acceptable for publication in the IJMS. We are looking forward to receiving your answer soon.

Sincerely,

Yeong-Min Yoo Ph.D 

Institute of Environmental Research,

Kangwon National University,

Chuncheon 24341, Republic of Korea

Email: yyeongm@hanmail.net

Reviewer 3 Report

Comments and Suggestions for Authors

The authors have presented a review describing initially the invididual actions of Mel and PDRN, and the proposed  combined use of them in various applications.

Although I find this review interesting and informational, I have a few comments.

1) In many places, there is a lack of references: line149-154; 159-163; 459-463; 628-632; 688-707

2) Particularly, the authors have quoted no reference for the mentioned limited co-administration of both agents. I wondered whether the current literature suggests any synergistic effects of using both of them, or rather, it is just additive effects. If there are no substantial studies to show that the two agents act synergistically, this review should be limited to discussing the two individually.

3)Fig.1 is rather plain and could be made more illustrative and colorful.

Author Response

May 30, 2025

Dear Reviewer 3,

Thank you for your letter and for the reviewers’ comments concerning our manuscript entitled " Comparative Analysis of Melatonin and Polydeoxyribonucleotide: Treatment Effects and Potential Synergistic Applicability."

We completely agree with your suggestions regarding our manuscript. The manuscript is completely revised as you and your colleague requested.

The corrections and revisions are as follows:

The authors have presented a review describing initially the individual actions of Mel and PDRN, and the proposed combined use of them in various applications.

Although I find this review interesting and informational, I have a few comments

  1. In many places, there is a lack of references: line149-154; 159-163; 459-463; 628-632; 688-707.
    Answer: Thank you very much for your good comments.

All references have been added to the blank section.

  1. Particularly, the authors have quoted no reference for the mentioned limited co-administration of both agents. I wondered whether the current literature suggests any synergistic effects of using both of them, or rather, it is just additive effects. If there are no substantial studies to show that the two agents act synergistically, this review should be limited to discussing the two individually.

Answer: Thank you very much for your good comments.

As recently as May 2025, a paper was published on the use of melatonin and PDRN in combination therapy to restore ovarian function. So I added the following:

Lines 640-642: Most recently, the use of a bioactive scaffold of mesenchymal progenitor cells in combination with PDRN and melatonin to improve ovarian function was announced, with the potential to offer therapeutic benefits [186]. Therefore, the theoretical combination….

  1. Kim, D.S.; Jeong, D.S.; Park, S.Y.; Jung, J.W.; Lee, J.E.; Lee, J.K.; Baek, S.W.; Lee, D.R.; Han, D.K. Ovarian Function Restoration with Biomimetic Scaffold Incorporating Angiogenic Molecules and Antioxidant in Chemotherapy-Induced Perimenopausal Model. Adv. Healthc. Mater. 2025, 14, e2403944. doi: 10.1002/adhm.202403944.

We have also rewritten the conclusion as follows:

  1. Conclusion

5.1. Potential Synergistic Effects

5.2. Clinical Applications

5.3. Future Directions

5.4. Summary

Lines 717-734: This paper highlights the promising therapeutic potential of combining melatonin and PDRN, leveraging their distinct pharmacological properties. Melatonin, known for modulating sleep, oxidative stress, and inflammation, complements PDRN's ability to promote tissue regeneration and angiogenesis. Individually, they offer significant benefits; however, their combined anti-inflammatory and antioxidant effects are proposed to profoundly enhance healing processes across various tissues, particularly in conditions marked by inflammation, oxidative stress, and tissue degeneration.

Given these analytical considerations and existing research, co-administering melatonin and PDRN may provide a synergistic approach to managing oxidative stress and tissue damage, with broad implications for wound healing and beyond (Figure 1).

While the combination shows considerable promise, extensive empirical validation through comprehensive clinical trials is paramount. These studies are not just beneficial, they're crucial for developing standardized treatment protocols that fully unlock the therapeutic potential of these agents. Future research must optimize these combination therapies to definitively confirm that their complementary mechanisms offer enhanced benefits over individual uses. Ultimately, this review underscores the urgent need for in-depth research to fully understand their synergistic effects and establish effective therapeutic strategies across diverse medical fields.

  1. Fig.1 is rather plain and could be made more illustrative and colorful.

Answer: Thank you very much for your good comments.

Figure 1 is reproduced again.

**All of the edited sections and references were changed with the blue words.

***The Certificate of Editing is attached.

I hope that the revised manuscript is now acceptable for publication in the IJMS. We are looking forward to receiving your answer soon.

Sincerely,

Yeong-Min Yoo Ph.D 

Institute of Environmental Research,

Kangwon National University,

Chuncheon 24341, Republic of Korea

Email: yyeongm@hanmail.net

Round 2

Reviewer 1 Report

Comments and Suggestions for Authors

Thanks for author’s responses. However, Conclusions section is too long, which should be refined and not include several subsections.

Author Response

June 2, 2025

Dear Reviewer 1,

Thank you for your letter and for the reviewers’ comments concerning our manuscript entitled " Comparative Analysis of Melatonin and Polydeoxyribonucleotide: Possible Benefits of Co-treatment Effects and Potential Synergistic Applicability."

We completely agree with your suggestions regarding our manuscript. The manuscript is completely revised as you and your colleague requested.

The corrections and revisions are as follows:

  1. Thanks for author’s responses. However, Conclusions section is too long, which should be refined and not include several subsections.

Answer: Thank you very much for your good comments.

We replaced it with the following:

  1. Comprehensive research framework

5.1. Potential Synergistic Effects

5.2. Potential Clinical Applications

5.3. Future Directions

  1. Conclusion

The English could be improved to more clearly express the research.

Answer: Thank you very much for your good comments.

The Certificate of Editing is attached.

**All of the edited sections were changed with the blue words.

***The Certificate of Editing is attached.

I hope that the revised manuscript is now acceptable for publication in the IJMS. We are looking forward to receiving your answer soon.

Sincerely,

Yeong-Min Yoo Ph.D. 

Institute of Environmental Research,

Kangwon National University,

Chuncheon 24341, Republic of Korea

Email: yyeongm@hanmail.net

Reviewer 3 Report

Comments and Suggestions for Authors

The authors have replied to my comments. However, about the potential synergism, I still have the concern on the discussion of ref. 186. The authors have not discussed the details of the results. If there is synergism, please state so; if not, please be convervative in saying that the synergism is still a hypothesis. Thus, the Title of the MS should not be an overstatement, and should be plainly as ".... possible benefits of co-treatment of...."

Author Response

June 2, 2025

Dear Reviewer 3,

Thank you for your letter and for the reviewers’ comments concerning our manuscript entitled " Comparative Analysis of Melatonin and Polydeoxyribonucleotide: Possible Benefits of Co-treatment Effects and Potential Synergistic Applicability."

We completely agree with your suggestions regarding our manuscript. The manuscript is completely revised as you and your colleague requested.

The corrections and revisions are as follows:

The authors have replied to my comments. However, about the potential synergism, I still have the concern on the discussion of ref. 186. The authors have not discussed the details of the results. If there is synergism, please state so; if not, please be conservative in saying that the synergism is still a hypothesis. Thus, the Title of the MS should not be an overstatement, and should be plainly as ".... possible benefits of co-treatment of...."
Answer: Thank you very much for your good comments.

We really appreciate the reviewer's suggestions. The synergistic effect of melatonin and PDRN co-treatment has not yet been demonstrated as suggested by the reviewer. We are currently working on it and will submit it as a research paper as soon as the results are available.

We have changed the title as suggested by the reviewer: Comparative Analysis of Melatonin and Polydeoxyribonucleotide: Possible Benefits of Co-treatment Effects and Potential Synergistic Applicability.

Lines 30-32: This review emphasizes the importance of exploring their potential synergistic effects and developing effective therapeutic strategies across various medical disciplines.

  1. Comprehensive research framework

5.1. Potential Synergistic Effects

5.2. Potential Clinical Applications

5.3. Future Directions

  1. Conclusion

**All of the edited sections were changed with the blue words.

I hope that the revised manuscript is now acceptable for publication in the IJMS. We are looking forward to receiving your answer soon.

Sincerely,

Yeong-Min Yoo Ph.D.

Institute of Environmental Research,

Kangwon National University,

Chuncheon 24341, Republic of Korea

Email: yyeongm@hanmail.net